# Differences in mediolateral dynamic stability during gait initiation according to whether the non-paretic or paretic leg is used as the leading limb

Yuji Osada[1]*, Naoyuki Motojima[2], Yousuke Kobayashi[3], Sumiko Yamamoto[4]

1 Department of Health and Welfare Tokushima Bunri University, Tokushima, Japan, 2 Showa University School of Nursing and rehabilitation Science, Kanagawa, Japan, 3 Nakaizu Rehabilitation Center, Shizuoka, Japan, 4 Graduate School, International University of Health & Welfare, Tokyo, Japan

* osada@tks.bunri-u.ac.jp

**Data Availability Statement:** All relevant data within the paper and its Supporting Information files.

## Abstract

We investigated mediolateral dynamic stability at first foot off and first initial contact during gait initiation according to whether the paretic or non-paretic leg was used as the leading limb. Thirty-eight individuals with stroke initiated gait with the paretic and non-paretic legs as the leading limb, and their movements were measured using a 3D motion analysis system. Margin of stability (i.e., the length between the extrapolated center of mass and lateral border of the stance foot) was used as an index of dynamic stability, with a large value indicating dynamic stability in the lateral direction. However, an excessively large margin of stability value (i.e., when the extrapolated center of mass is outside the medial border of the stance foot) indicates dynamic instability in the medial direction. Differences in the margin of stability between tasks were compared using the Wilcoxon signed-rank test. The minimum margin of stability was observed just before first foot off. When the non-paretic leg was used as the leading limb, the margin of stability tended to be excessively large at first foot off compared with when the paretic leg was used (p < 0.001). In other words, the extrapolated center of mass was outside the medial border of the paretic stance foot. In conclusion, lateral stability was achieved when using the non-paretic leading limb because the extrapolated center of mass was located outside the medial border of the stance foot. However, medial dynamic stability was lower for the non-paretic leading limb compared with the paretic leading limb.

## Introduction

Posture is unconsciously controlled during steady-state walking but must be voluntarily controlled at gait initiation (GI) [1]. The cerebral cortex is activated before the floor reaction force changes in the GI task [2], and the basal ganglia, thalamus, and cerebellum are also involved at the start of movement [3]. Accordingly, various problems occur in patients with stroke, such as abnormal muscle activity [4] and delayed muscle activation [5]. The GI task is challenging in terms of the stability of control dynamics, particularly in the mediolateral direction [6]. In

**Funding:** The authors received no specific funding for this work.

**Competing interests:** The authors have declared that no competing interests exist.

individuals with stroke, the paretic lower limb can be used as either the leading or trailing limb at GI. The leading limb is responsible for the initial transfer of weight to the stance limb as well as the antigravity flexion required for forward swing [7]. The trailing limb, on the other hand, is responsible for antigravity extension, which is required to support body weight and generate forward momentum for the first step [7]. Thus, physical therapists sometimes need to confirm that individuals have selected an effective leading limb, especially individuals with stroke.

Although evaluation of mediolateral stability is important for GI [8], few studies have analyzed differences in the selection of the leading limb from the viewpoint of stability. A recent systematic review argued against use of the paretic leading limb because it provides poor balance, noting also that the non-paretic leading limb stimulates the paretic postural muscles [4]. However, the same article also stated that use of the non-paretic leg as the leading limb poses a greater challenge in terms of balance during GI because the weakness of the paretic leg makes it difficult to support the body weight during the subsequent stance phase. A previous study comparing the change in the center of pressure (CoP), stride, and duration in 14 participants with stroke also recommended use of the paretic leading limb because the starting posture is asymmetrical and the first step length is shortened when the non-paretic leg is used as the leading limb [9]. The other previous study comparing ground reaction forces and GI speed in 13 participants with stroke recommended use of the non-paretic leading limb because, when subjects with stroke initiated with their non-paretic limb, the anteroposterior force and impulse generated by the trailing limb were strongly related to the magnitude of the final GI speed [10]. The reasons for these disagreements among the previous studies are as follows: i) few previous studies have investigated the selection of the leading limb, ii) different evaluation indices were used in the studies, and iii) the number of participants was small. The latter factor might have undermined the statistical power of the analyses.

Range of CoP movement is the classical method for analyzing postural stability [11,12]. The range of CoP movement is not a quantitative index for evaluating stability because the CoP does not move outside the base of support, even when balance is lost, and the range of the CoP movement is limited. A previous study [13] proposed the extrapolated center of mass (Xcom) as an index of dynamic motion stability based on use of an inverted pendulum model to calculate the margin of stability (MoS). In this study, the main outcome was MoS in the mediolateral direction, because mediolateral dynamic stability at first stance is the most important aspect for individuals with hemiplegia [14]. The aim of this study was to clarify the advantages and disadvantages of the use of the paretic and non-paretic legs as the leading limb from the viewpoint of mediolateral dynamic stability in individuals with stroke. Additionally, we sought to perform this study in a larger sample of patients than in previous studies. Previous work [9] has shown instability associated with the use of the non-paretic leading limb from the viewpoint of CoP movement. However, another study that analyzed the velocity of the center of mass indicated that use of the non-paretic leading limb was advantageous [10]. The leading limb also plays a role in predictive postural control [7], for which the non-paretic lower limb may be more effective. Therefore, here we examined the hypothesis that use of the non-paretic leg as the leading limb would be more stable from the viewpoint of dynamic stability in individuals with stroke compared with use of the non-paretic leg. Clarification of whether the choice of leading limb affects dynamic stability would provide information on the appropriate first step for individuals with stroke.

## Materials and methods

### Participants

A total of 38 individuals with stroke (mean age, 59.5 ± 10.1 years) participated in this cross-sectional study. Participant characteristics are shown in Table 1. All participants were patients in

**Table 1. Participant characteristics (N = 38).**

| | |
|---|---|
| Age [years], mean ± SD | 59.5 ± 10.1 |
| Weight [kg], mean ± SD | 59.2 ± 11.0 |
| Height [cm], mean ± SD | 163.9 ± 7.4 |
| Sex (female/male) | 11/27 |
| Paretic side (right/left) | 17/21 |
| Time since onset (days), median (IQR) | 92 (98) |
| BRS: lower extremity score, n (%) | III: 7 (18); IV: 9 (24); V: 18 (47); VI: 4 (11) |
| FMA: lower extremity score, median (IQR) | 28 (3.75) |
| FMA: balance score, median (IQR) | 10 (3.75) |
| FIM: transfer to the bed score, n (%) | V: 4 (13); VI: 18 (47); VII: 16 (42) |
| FIM: locomotion (walking) score, n (%) | IV: 4 (11); V: 10 (26); VI: 15 (39); VII: 9 (24) |
| 10-m timed walk test (s), mean ± SD | 20.7 ± 14.7 |

BRS, Brunnstrom recovery stage; FMA, Fugl–Meyer assessment; FIM, Functional Independence Measure; IQR, interquartile range.

a convalescent rehabilitation ward who met the following inclusion criteria: (1) hemiparesis secondary to cerebrovascular accident; (2) first unilateral stroke; and (3) able to follow simple instructions and walk at least 10 m at their preferred speed without manual assistance. Individuals were excluded if they had other neurological or musculoskeletal deficits that would affect gait. This study was approved by the ethics committees of International University of Health & Welfare (17-Io-22) and Nakaizu Rehabilitation Center (28–007). Participants provided written informed consent before enrollment in the study.

## Study protocol

During two tasks (i.e., use of the paretic leading limb and use of the non-paretic leading limb at GI), data were measured using a 3D motion capture system comprising eight Vicon MX cameras (Vicon Motion System Ltd., Oxford, UK) and six AMTI force plates (600 mm × 400 mm; Advanced Mechanical Technology Inc., Watertown, MA). Participants were positioned in a standardized static standing position with each foot placed on two separate force plates at pelvis width [15] because the wider the initial stance width, the greater the mediolateral movement [16]. Mediolateral stability changes according to the initial speed [8]. Thus, to ensure that the GI conditions were uniform for all participants, they were instructed to start walking as quickly as possible from a static standing position toward a line 3 m ahead [17] upon receiving a cue from a buzzer synchronized with an LED light (Fig 1). Following a reported protocol [9,18], participants wore normal low-heeled shoes; no ankle-foot orthosis or walking aid was used. First, participants repeated the GI trial three times with no instruction on which limb should be used as the leading limb; then, the participants were instructed to use the other leading limb for three trials. This procedure was chosen because random determination of the leading limb tended to cause the patients to freeze and behave unnaturally. Guarded assistance from physiotherapists during the task prevented the participants from falling.

A total of 34 reflective markers were attached to the participants at various landmarks, following a protocol used in previous studies [19,20]. The marker trajectories and force plate data were synchronized at a sampling frequency of 120 Hz. Marker trajectories and force plate data were low-pass filtered using a second-order Butterworth filter with cutoff values of 6 Hz and 18 Hz, respectively [21]. Center of mass and joint centers were calculated using anthropometric data [22] for the following 12 link-segment model: feet (head of the second metatarsal joint,

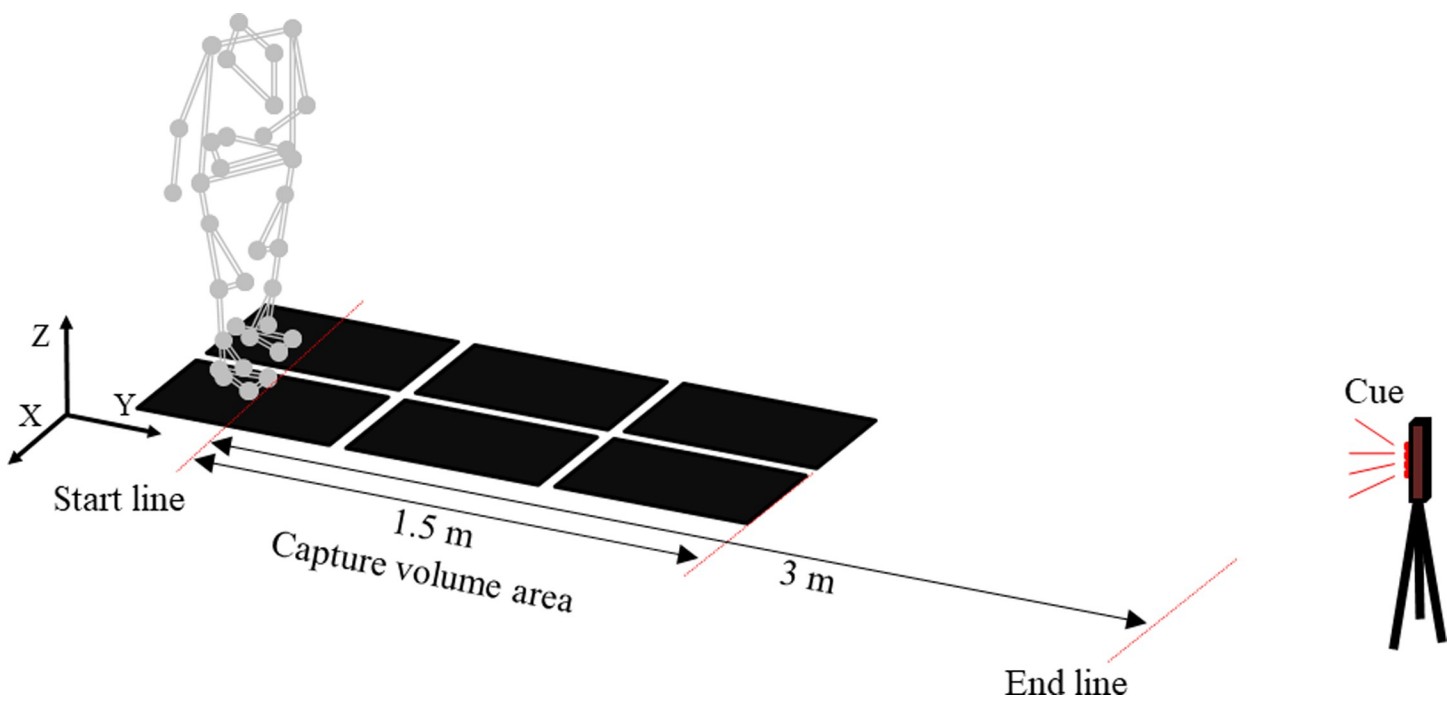

**Fig 1. Experimental set-up for GI with 34 reflective markers and 6 force plates.**

fifth metatarsal joint, and heel markers), shanks (lateral malleolus, medial malleolus, and lateral femoral epicondyle markers), thighs (lateral femoral epicondyle, medial femoral epicondyle, and hip joint markers), pelvis (bilateral anterior superior iliac spine and posterior superior iliac spine markers), upper trunk (by T2 and T10 vertebrae, sternal notch, and jugular notch markers), upper arms (acromion and elbow lateral markers), and forearms (elbow lateral and wrist lateral markers). Joint kinematics and kinetics were calculated using an inverse dynamic model according to the Vicon Plug-in Gait model. Kinetic and kinematic data were exported into a biomechanics analysis program (Visual3D, version 5; C-Motion, Inc., Germantown, MD).

## Parameters

The movement task was separated into three phases [23]: 1) the postural phase, from onset to first foot off; 2) the monopodal phase, from first foot off to first initial contact; and 3) the double support phase, from first initial contact to second foot off. Onset was defined as the start of the reverse reaction phenomenon of the CoP, namely, when the vertical component of the floor reaction force on the leading limb begins to continuously increase. To analyze the differences in dynamic stability at GI between each leading limb, we extracted the MoS, which is an index of dynamic motion stability at GI [8,24]. A previous study [13] proposed the use of Xcom, which is based on use of the inverted pendulum model to calculate the MoS. Using the formula from previous studies [13,25], the mediolateral MoS was calculated as the distance between Xcom and the boundaries of the base of support (BoS) during single stance from the foot markers, namely, the fifth metatarsal-phalangeal (MP) joint for the lateral border and the medial malleolus for the medial border. A visual representation of the MoS calculation is shown in Fig 2. The MoS can be calculated in medial and lateral directions and %MoS was calculated as the ratio of the distance (%) from the BoS boundary to Xcom, with the distance

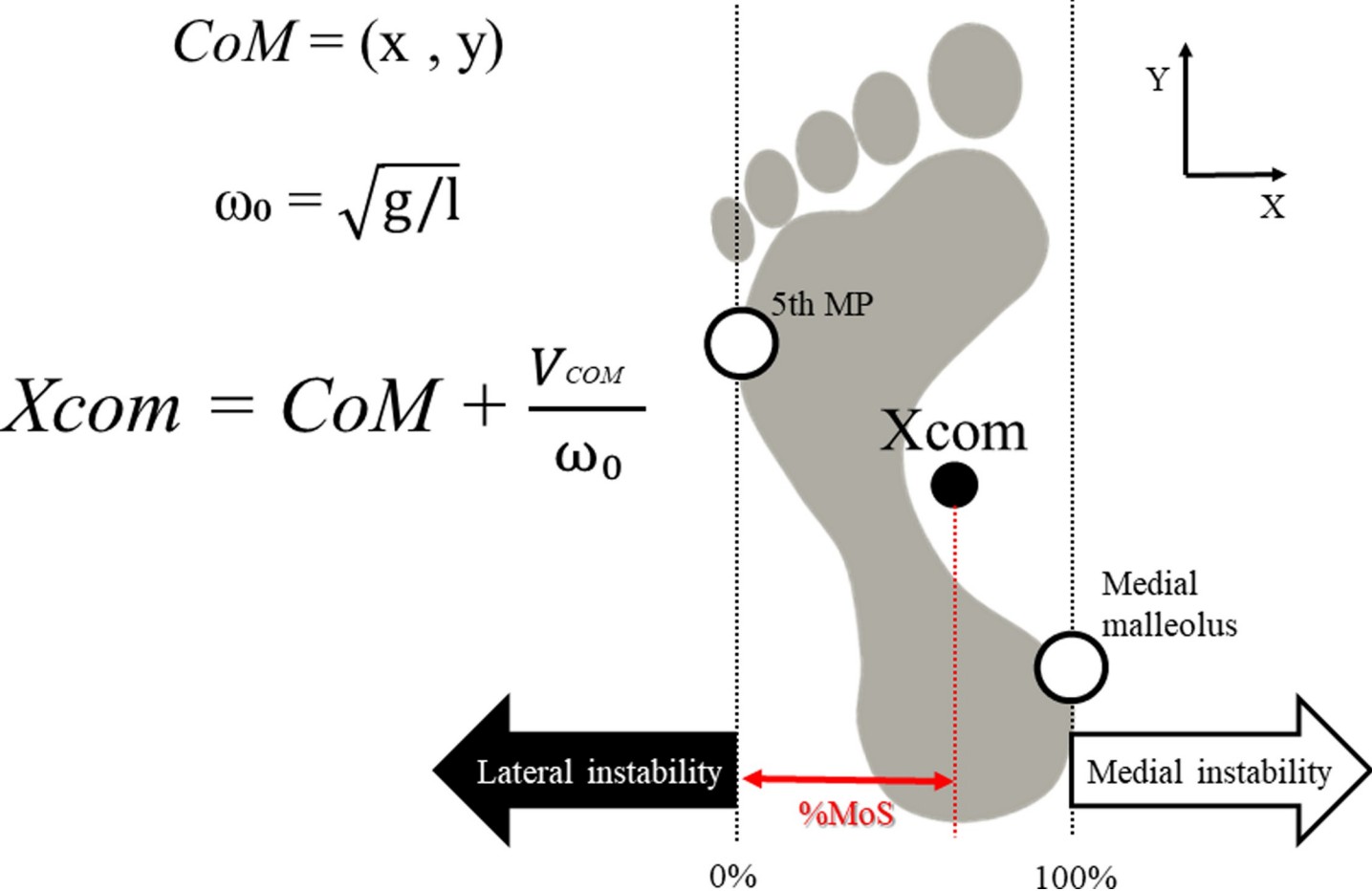

**Fig 2. Illustration of the MoS calculation at foot off.** The black circle indicates the location of Xcom while the white circles indicate key markers of the BoS boundary in the mediolateral direction. %MoS was calculated as the ratio of the distance (%) from the BoS boundary to Xcom, with the distance from the 5th MP to the medial malleolus being 100%. Abbreviations: 5th MP, fifth metatarsal-phalangeal joint; $\omega_0$, the eigenfrequency of the inverted pendulum model; CoM, center of mass; g, gravity; l, length of the leg; MoS, margin of stability; Vcom, velocity of the CoM; Xcom, extrapolated center of mass.

from the 5th MP to the medial malleolus taken as 100%.

$$\%Mos = \frac{Xcom - 5th\ MP}{medial\ malleolus - 5th\ MP} \times 100$$

The values of %MoS were extracted at first foot off (%MoS_FO) and initial contact (%MoS_IC), as in a previous study [26]. A %MoS value lower than 0% indicates decreased dynamic stability in the lateral direction, while a larger %MoS indicates greater lateral stability. However, a %MoS value above 100% indicates that Xcom is inside the stance foot. Therefore, only a %MoS value between 0% and 100% would indicate stability in single support. Because stability in the lateral direction is the most important factor in patients with stroke [14], the primary index for determining stability is "MoS is a large value" and the secondary index is "MoS does not exceed 100%".

The following additional indices were also used to compare the advantages and disadvantages of the leading limb selection, as in previous work [4,9]: movement duration, step length, and lateral pelvic tilt angle. Movement duration was calculated for each of three phases: 1) the postural phase, 2) monopodal phase, and 3) double support phase. In previous studies, these

indicators were used to determine the advantage of the leading limb selection from the view-point of movement efficiency (duration) or posture symmetry (step length or lateral pelvic tilt). Step length was calculated as the anteroposterior distance between the heel markers at initial contacts. The lateral pelvic tilt was calculated based on changes from the static standing position to first foot off because the pelvis is lifted most at first foot off in individuals with stroke [27].

GI tasks were measured without the use of walking aids, even in those who usually use canes (17 persons) and orthoses (13 persons). Accordingly, in patients who use walking aids, the GI movement might have a different pattern. Therefore, as an additional analysis, the participants were divided into two groups (walking aid user or not), and the differences in %MoS-FO and %MoS-IC were compared in each group between the tasks performed using paretic and non-paretic leading limbs.

## Statistical analysis

Differences in %MoS, movement duration, step length, and lateral pelvic tilt were compared between the tasks performed using the paretic and non-paretic leading limbs. All indices were extracted as the average of three trials for each task. The Shapiro–Wilk test was used to assess the normality of all data. Normally distributed data were compared using the paired *t*-test, whereas non-normally distributed data were compared using the Wilcoxon signed-rank test (SPSS, version 24.0; IBM Corp., Armonk, NY). The significance level was set at $\alpha = 0.05$, and the effect size *r* [28] was calculated for each index. According to Cohen [29], a large effect is represented by an *r* of at least 0.50, a moderate effect by 0.30, and a small effect by 0.10. The formula for calculating the effect size *r* was

$$r = \frac{Z}{\sqrt{n}}$$

where *Z* is the Z score on the Wilcoxon signed-rank test and *n* is the number of pairs.

## Results

In the first three trials, 17 participants used the paretic leg consecutively as the leading limb and 6 used the non-paretic leg. Fifteen participants did not select a specific leading limb in consecutive trials. The differences in the average %MoS displacements in the mediolateral direction are shown in Fig 3. No participants had negative %MoS values in the mediolateral direction. Regardless of which limb was used as the leading limb at GI, %MoS showed its minimum value just before the first foot off, namely, Xcom moved the most in the lateral direction at this time. In addition, in most participants, %MoS-FO tended to be from 0% to 100% when the paretic leg was used as the leading limb. When the non-paretic leg was used as the leading limb (solid line), Xcom tended to move less toward the paretic foot and the absolute minimum %MoS value exceeded 100% in 20 of the 38 participants, that is, Xcom did not reach the BoS comprising the paretic lower limb. When the paretic leg was used as the leading limb (dotted line), the absolute minimum %MoS value exceeded 100% in only 8 of the 38 participants.

Differences in the mediolateral %MoS, movement duration, step length, and lateral pelvic tilt angle are shown in Table 2. The results showed significant differences in %MoS-FO and %MoS-IC with a large effect size (r = 0.82, P < 0.001). The median difference between tasks was 24%. When the non-paretic leg was used as the leading limb, %MoS-FO and %MoS-IC were larger compared with when the paretic leading limb was used. Regarding the movement duration, when the non-paretic leading limb was used, the postural phase was longer and the monopodal support phase was 0.13 seconds shorter than when the paretic leading limb was used (effect size r = −0.66, P < 0.001). The pelvic lateral tilt angle reached its maximum value

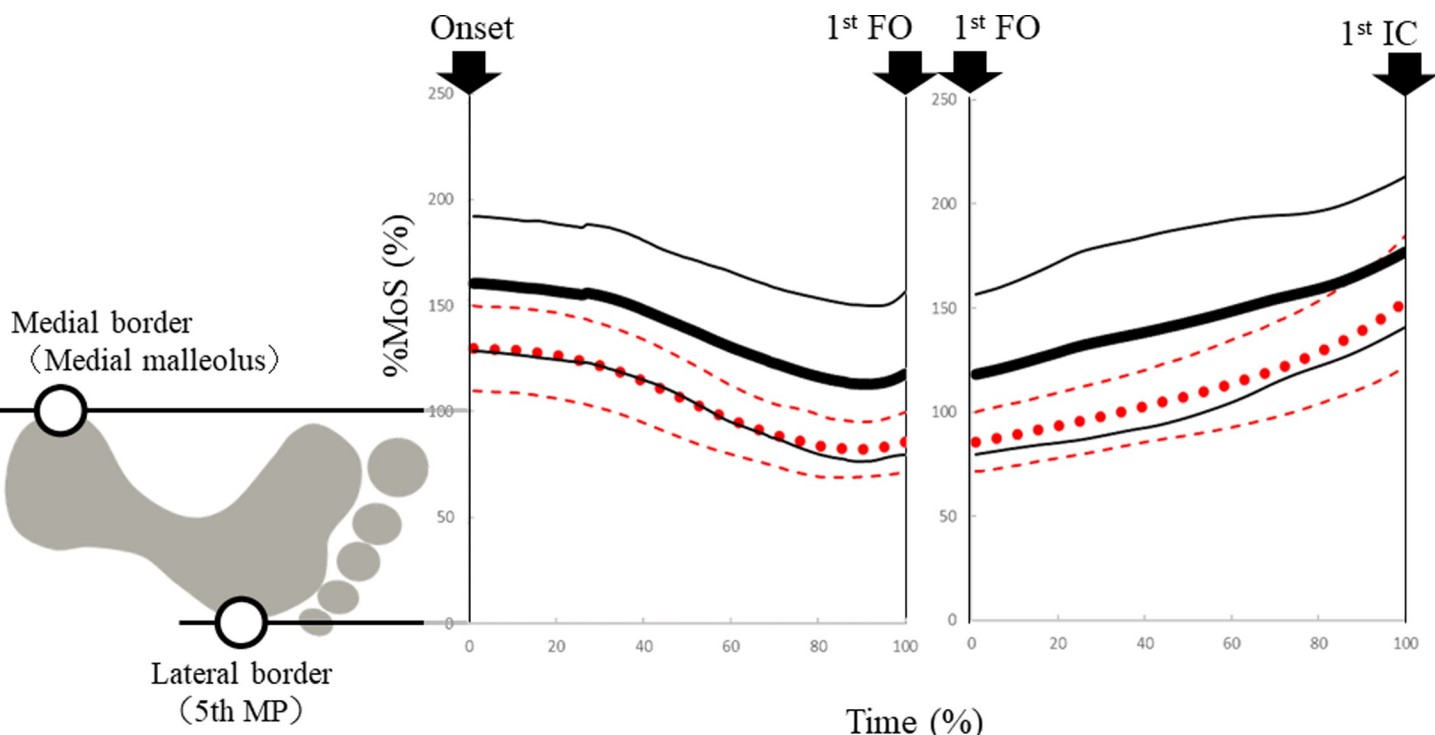

**Fig 3. Differences in the average %MoS between tasks performed using paretic and non-paretic leading limbs (n = 38).** The solid line indicates the task performed using the non-paretic leading limb. The dotted line indicates the task performed using the paretic leading limb. Dashed lines around the dotted line for the paretic leg and fine solid lines for the non-paretic leg indicate the standard deviation. The trajectories were obtained for the averaged trial of each subject with GI begun as quickly as possible from a static standing position. The horizontal axis is time (%), which is normalized to 100% each from onset to first foot off and from first foot off to first initial contact. The vertical axis is the distance (%) from the lateral boundary of the BoS to Xcom. The illustration of the foot is an image of the mediolateral position of the first stance foot. Mediolateral BoS (5th MP to medial malleolus) is normalized to 100, with negative values indicating instability in the lateral direction. Abbreviations: 5th MP, fifth metatarsal-phalangeal joint; BoS, base of support; CoM, center of mass; FO, foot off; IC, initial contact; MoS, margin of stability; Xcom, extrapolated center of mass.

at first foot off. When the non-paretic leg was used as the leading limb, the lateral pelvic tilt was 2.8˚ smaller at first foot off compared with when the paretic leading limb was used (effect

**Table 2. Differences between use of the paretic and non-paretic leg as the leading limb (N = 38).**

| | Paretic leading limb | Non-paretic leading limb | *p*-value | Effect size (*r*) |
|---|---|---|---|---|
| %MoS (%) | | | | |
| %MoS-FO#, median (IQR) | 86.1 (22.8) | 110.8 (34.5) | <0.001 | 0.82 |
| %MoS-IC#, median (IQR) | 155.0 (28.0) | 172.8 (50.1) | 0.008 | 0.50 |
| Duration (s) | | | | |
| Postural phase#, median (IQR) | 0.18 (0.10) | 0.20 (0.18) | 0.014 | 0.40 |
| Monopodal phase#, median (IQR) | 0.43 (0.19) | 0.30 (0.10) | <0.001 | −0.66 |
| Double support phase#, median (IQR) | 0.28 (0.22) | 0.27 (0.20) | 0.294 | 0.17 |
| Step length (cm/Ht), mean ± SD | | | | |
| First step | 0.199 ± 0.077 | 0.208 ± 0.077 | 0.290 | 0.17 |
| Second step | 0.182 ± 0.095 | 0.189 ± 0.081 | 0.419 | 0.13 |
| Lateral pelvic tilt angle at first FO (deg)#, median (IQR) | 4.14 (3.30) | 1.34 (4.51) | <0.001 | 0.76 |

#Non-normally distributed; %MoS-FO, %MoS at first foot off; %MoS-IC, %MoS at first initial contact; Ht, height; IQR, interquartile range; MoS, margin of stability; postural phase, onset to first foot off; monopodal phase, first foot off to first initial contact; and double support phase, first initial contact to second foot off.

size r = 0.76, P < 0.001). In additional analysis that divided the participants into two groups (walking aid user or not), %MoS$_{-FO}$ and %MoS$_{-IC}$ showed similar results and there were significant differences only in walking aid users (n = 20). The non-walking aid users (n = 18) had no significant differences in %MoS$_{-FO}$ between use of the paretic and non-paretic leading limb. However, they had the same tendency in the above result; that is, %MoS$_{-FO}$ exceeded 100% when a non-paretic leg was used as the leading limb.

## Discussion

This is the first study to clarify which leading limb is more stable from the viewpoint of mediolateral dynamic stability in individuals with stroke. When the non-paretic leg was used as the leading limb, %MoS-FO and %MoS-IC were larger in the medial direction at GI compared with when the paretic leading limb was used. This result indicates dynamic stability in the lateral direction with the non-paretic leading limb. However, regarding the medial direction, this result was contrary to our hypothesis that use of the non-paretic leg as the leading limb is more stable for individuals with stroke for GI than use of the paretic leg as the leading limb.

GI poses a challenge in terms of mediolateral dynamic stability in individuals with stroke [4,8,18] because they have various problems associated with the transfer of weight onto the paretic limb, such as sensory loss, muscle weakness, and fear. Individuals with stroke tend to adopt an asymmetric posture by reducing the load on the paretic limb not only during static standing, but also for steps and when walking [9,30,31]. In this study, the monopodal phase was shortened when the non-paretic leading limb was used. This is also one of the compensatory strategies to reduce loading on the paretic limb. Other previous studies of GI have also mentioned this tendency after stroke in the subacute phase [9] and chronic phase [10]. In this study, %MoS$_{-FO}$ and %MoS$_{-IC}$ exceeded 100% in more than half of the participants when the non-paretic limb was used as the leading limb. Thus, these negative behaviors toward paretic limb loading may have caused an excessively large MoS when the non-paretic leading limb was used, that is, when the paretic leg was used as the first stance limb.

In a previous study comparing MoS in the lateral direction and various balance indices in individuals with stroke, clinical balance indices had a negative correlation with MoS in the lateral direction of the paretic leg [32]. This previous research mentioned that maintenance of a large MoS in the lateral direction indicated a poor ability to maintain balance. Therefore, a large MoS in the paretic direction indicates that they might intentionally limit the center of mass transfer on the paretic side. This compensatory movement to focus on instantaneous balance is the reason why %MoS exceeded 100% when the non-paretic leg was used as the leading limb (i.e., as the first stance phase of the paretic limb).

In single support, only a %MoS value between 0% and 100% indicates stability. Therefore, especially for dynamic stability in the lateral direction, we conclude that use of the non-paretic leading limb results in higher dynamic stability than use of the paretic leading limb. However, especially for dynamic stability in the medial direction, we conclude that use of the non-paretic leading limb results in worse dynamic stability than use of the paretic leading limb. Patients with stroke may intentionally maintain such an excessive %MoS to avoid lateral perturbation, which leads to falls. The excessive lateral stability might not lead to a feeling of reassurance in individuals after stroke, even though the kinematic stability of the lateral direction was guaranteed, because the fact that the MoS persistently exceeded 100% suggests instability in the medial direction, with participants forced to immediately take another step. Some previous studies reported that adults and children with hemiplegia tend to use a paretic leading limb because of their asymmetric standing posture, which is the result of less limb loading on the paretic side [30,33]. The postural phase was shortened in our study, with the first foot off earlier with the

paretic leading limb than with the non-paretic leading limb. This result may indicate that individuals with stroke may find it easier to lift the paretic lower limb as the leading limb.

Regarding other indices, there was no difference in step length, regardless of the leading limb selection, which is in contrast to the result of a previous study [9]. However, there was a significant difference in the lateral pelvic tilt at first foot off, which showed a large value when the paretic leg was used as the leading limb. This result is similar to that of Davies et al. [27], who found that, if patients start walking using the paretic leading limb, the pelvic is excessively actively lifted. This seems to be compensatory hip hiking for toe clearance [34]. The pelvic tilt angle when the non-affected leading limb was used (median, 1.34°) was similar to that in healthy individuals (average, 0 ± 1°) [35]. However, the angle when the paretic leading limb was used (median, 4.14°) was three times larger than that when the non-paretic leading limb was used. The minimal clinically important difference (MCID) for the pelvic elevation angle has not been determined. Nonetheless, it can be said that GI deviates asymmetrically from normal movement when the paretic leg is used as the leading limb.

Given the above, we can conclude that use of the non-paretic leading limb is disadvantageous for mediolateral stability because %MoS$_{-FO}$ and %MoS$_{-IC}$ values exceeding 100% may indicate instability, unless the patient steps rapidly in the medial direction. However, there is an advantage in postural symmetry. This insufficient weight transfer to the paretic limb can represent compensatory movement to obtain biomechanical stability when the non-paretic leading limb is used. Instruction of individuals with stroke to use the paretic leg as the leading limb can be useful from the viewpoint of mediolateral dynamic stability. However, it may be necessary to suggest use of the non-paretic limb as the leading limb to individuals with stroke and high balance function to improve postural symmetry because the non-walking aid users had no significant differences in %MoS$_{-FO}$ between use of the paretic and non-paretic leading limbs.

There are several limitations to this study. First, because it was conducted with limited evaluation indices, it was not possible to analyze the results based on the participants' physical function, use of a walking aid, dominant foot, and favored leading limb. Second, we focused on only kinematic indicators, and other factors related to subjective impressions may also need to be considered regarding leading limb selection. Third, to eliminate the effect of walking aids, GI tasks were measured without the use of walking aids, even in those who usually use canes (17 persons) and orthoses (13 persons). Accordingly, if the patients use walking aids, the GI movement might have different patterns. Despite these limitations, few other studies have examined the selection of the leading limb at the initiation of walking in a large patient sample and the results of this study could thus be useful for the development of rehabilitation programs focusing on movements at walking initiation.

## Conclusions

This study examined the mediolateral dynamic stability in GI from the viewpoint of differences in the selection of the leading limb in individuals with stroke. When poststroke individuals used the non-paretic leg as the leading limb, the MoS in the lateral direction was excessively large compared with when the paretic limb was used. Therefore, medial dynamic stability was lower when individuals with stroke initiated walking with the non-paretic leg than with the paretic leg. These results should help physical therapists when they are recommending the appropriate first step for individuals with stroke.

## Supporting information

**S1 File.**
(XLSX)

## Author Contributions

**Conceptualization:** Yuji Osada.

**Data curation:** Yuji Osada, Yousuke Kobayashi.

**Formal analysis:** Yuji Osada.

**Investigation:** Yuji Osada.

**Methodology:** Yuji Osada.

**Project administration:** Yuji Osada.

**Resources:** Yuji Osada.

**Software:** Yuji Osada.

**Supervision:** Naoyuki Motojima, Sumiko Yamamoto.

**Validation:** Yuji Osada.

**Visualization:** Yuji Osada.

**Writing – original draft:** Yuji Osada.

**Writing – review & editing:** Naoyuki Motojima, Sumiko Yamamoto.

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
