## [Decision Letter · Decision Letter 0]

13 Jan 2022

PONE-D-21-29688Differences in mediolateral dynamic stability during gait initiation according to whether the non-paretic or paretic leg is used as the leading limbPLOS ONE

Dear Dr. Osada,

Thank you for submitting your manuscript to PLOS ONE. After careful consideration, we feel that it has merit but does not fully meet PLOS ONE’s publication criteria as it currently stands. Therefore, we invite you to submit a revised version of the manuscript that addresses the points raised during the review process. Both reviewers commented on the interpretation of the results of the study. This needs to be improved before the paper can be considered for publication. Several other issues also require attention.

We look forward to receiving your revised manuscript.

Kind regards,

Peter Andreas Federolf

Academic Editor

PLOS ONE

Journal Requirements:

Reviewers' comments:

Reviewer's Responses to Questions

**Comments to the Author**

1. Is the manuscript technically sound, and do the data support the conclusions?

Reviewer #1: Partly

Reviewer #2: Partly

2. Has the statistical analysis been performed appropriately and rigorously? 

Reviewer #1: Yes

Reviewer #2: Yes

3. Have the authors made all data underlying the findings in their manuscript fully available?

Reviewer #1: No

Reviewer #2: Yes

4. Is the manuscript presented in an intelligible fashion and written in standard English?

Reviewer #1: Yes

Reviewer #2: Yes

5. Review Comments to the Author

Reviewer #1: This study investigated the mechanics of gait initiation among patients post stroke, comparing their gait initiation with the paretic vs their non-paretic leg. The study is highly relevant for stroke rehabilitation in practice and adds value to understanding gait mechanisms post-stroke. However, there are certain methodological issues that will need to be addressed before being eligible for publication.

The main issue with this paper is the interpretation of the results leading to the main conclusion; differences are found in %MoS during GI, with a lower %MoS when the paretic leg is leading. This finding led to the conclusion that using the non-paretic leg as the leading limb is more stable compared to using the paretic leg, however, alternative conclusions are possible from this finding. That is, when the paretic leg led, movement times in the monopodal phase were significantly longer and in single support the interpretation of a higher %MoS as an indicator for stability is void. That is, in single support only a %MoS between 0 and 100 would indicate stability. The decreased values of %MoS could in this case indicate increased stability or a movement strategy more focused on instantaneous balance rather than quick progression.

A second major issue is that there is currently insufficient detail in the methods to reproduce the study. Especially in terms of the computation of the dependent variables, this should be described better. For instance, the following issues should be clarified:

• The protocol was performed three times per leading leg. How were these three trials analyzed, averaged or on a trial-by-trial level?

• The exact timing of sampling %MoS is unclear, for this, it would help to introduce the concepts of the paragraph on lines 127-135 earlier in the methods, perhaps supported by a figure.

• Related to the above point, in the results it is mentioned that “the %MoS showed its minimum value at first foot off” (line 152-153), whereas figure 3 shows this minimum to be just prior to first foot off.

• Finally related to this, in the results and discussion, the authors often mention %MoS without indicating which sampling moment is meant. I would suggest specifying with as %MoS-FO, %MoS-IC, or %MoS-both (or using subscript)

A final methodological issue lies in one of the factors that the authors also highlighted as a limitation; namely that all participants have been analyzed walking without aid, even though they might not be used to this. This analysis might not be sufficiently powered to lead to strong conclusions, but it would be helpful to add a subgroup analysis to get an indication of whether the main results are similar across different groups or driven by one group in particular.

Further minor issues need to be considered:

• The abstract is confusing when read without prior knowledge of the paper. For instance, terms as ‘first foot off’ should be more clearly described as well as a % margin of stability of over 100% (to be normalized to 100% implies that 100% is the maximum value), also it is not mentioned in which direction (medial or lateral) these values are directed.

• Line 39, posture is not always unconsciously controlled during walking and while it is true that gait initiation needs to be done voluntary, it could be debated to what extend posture is consciously controlled in this task.

• Line 45-47. “Thus, physical …” to “… with stroke.” This statement does not flow well. Why does a physical therapist determine this? Should patient not decide this?

• Line 59-60. Displacement needs to be defined better. Displacement cannot be defined as instability as the CoP is always moving, without always being instable. Furthermore, this neglects the functional role of CoP fluctuations (e.g. van Emmerik; van Wegen, doi:10.1097/00003677-200210000-00007).

• Line 66-67 & 75. Why was the number of 38 individuals chosen? Was a power calculation done based on the mentioned (but underpowered) previous studies?

• Line 85-107. How was a loss of balance prevented in this study? Did the participants wear a harness or was there a therapist nearby?

• Line 154-158. When discussing for how many participants %MoS exceeded 100%, was this measured on the two instances %MoS was sampled or the absolute minimal %MoS value for a participant?

• Line 158. “No participants had negative %MoS values in the mediolateral direction.” A negative %MoS would very quickly lead to a loss of stability and/or fall, which I assume would be prevented in the experimental design, to avoid injuries to the participant. Was it even possible to score a negative %MoS?

• Line 173. No hypothesis has been stated in the introduction.

• Figure 3. The standard deviation lines are not quite clear because of the added shading and their overlap. I would suggest taking away the shading and using a dotted line around the dashed line for the paretic leg and a fine solid line to indicate the standard deviation for the non-paretic leg.

Finally, I have highlighted elsewhere in the review system that I believe more could be done to adhere to the Data Availability policy of PLOS. I would like to further clarify this point here. I believe this is a very relevant study which collected data in a complex setting from a very select group. Sharing the raw data would not only increase the value of this article (improve reproducibility and allow for external validation), it would also hold ethical benefits as it would allow others to perform pilot analyses, without using valuable time and effort of this vulnerable patient group.

Reviewer #2: The data in the manuscript partly support the conclusions; however, the authors appear to overreach in the conclusions drawn from the margin of stability metric, which, although is a good measure of mediolateral postural stability, does not fully capture an individual's postural control during a task. Please see a complete list of detailed comments in the attached document.

6. PLOS authors have the option to publish the peer review history of their article (what does this mean?). If published, this will include your full peer review and any attached files.

Reviewer #1: No

Reviewer #2: No

---

## [Author Response · Author response to Decision Letter 0]

10 Feb 2022

Detailed Response to Reviewers.

Reviewer #1:

We wish to express our deep appreciation for your feedback. As indicated in our point-by-point responses below, we have taken all of your comments and suggestions into consideration in the revised manuscript. Major revisions are highlighted in yellow in the marked-up copy of the manuscript.

Comments

The main issue with this paper is the interpretation of the results leading to the main conclusion; differences are found in %MoS during GI, with a lower %MoS when the paretic leg is leading. This finding led to the conclusion that using the non-paretic leg as the leading limb is more stable compared to using the paretic leg, however, alternative conclusions are possible from this finding. That is, when the paretic leg led, movement times in the monopodal phase were significantly longer and in single support the interpretation of a higher %MoS as an indicator for stability is void. That is, in single support only a %MoS between 0 and 100 would indicate stability. The decreased values of %MoS could in this case indicate increased stability or a movement strategy more focused on instantaneous balance rather than quick progression.

Response:

Thank you for your feedback. In response to your advice, we have changed the interpretation of the results from “higher %MoS value indicate good stability” to “a %MoS exceeding 100% suggests instability.” Therefore, the conclusion was changed to “use of the non-paretic leading limb results in worse dynamic stability than use of the paretic leading limb. Patients with stroke may intentionally maintain such an excessive %MoS to avoid lateral perturbation, which leads to falls.” (lines 33–38, 290–293, 315–322, 346–357, 375–377).

A second major issue is that there is currently insufficient detail in the methods to reproduce the study. Especially in terms of the computation of the dependent variables, this should be described better. For instance, the following issues should be clarified:

• The protocol was performed three times per leading leg. How were these three trials analyzed, averaged or on a trial-by-trial level?

• The exact timing of sampling %MoS is unclear, for this, it would help to introduce the concepts of the paragraph on lines 127-135 earlier in the methods, perhaps supported by a figure.

• Related to the above point, in the results it is mentioned that “the %MoS showed its minimum value at first foot off” (line 152-153), whereas figure 3 shows this minimum to be just prior to first foot off.

• Finally related to this, in the results and discussion, the authors often mention %MoS without indicating which sampling moment is meant. I would suggest specifying with as %MoS-FO, %MoS-IC, or %MoS-both (or using subscript)

Response:

Thank you for your feedback. All indices were extracted for each task and then the average of three trials was used (line 217).

Regarding the exact timing of the %MoS sampling, we have defined the phase division of the movement task at the beginning of the Parameters section (lines 159–164). We extracted the feature values according to this phase classification.

The minimum %MoS value was found just before the first foot off. Accordingly, we have changed the sentence from “the %MoS showed its minimum value at the first foot off” to “the %MoS showed its minimum value just before the first foot off” (line 234). Additionally, the sampling moment was specified by using %MoS-FO and %MoS-IC, in response to your suggestion.

A final methodological issue lies in one of the factors that the authors also highlighted as a limitation; namely that all participants have been analyzed walking without aid, even though they might not be used to this. This analysis might not be sufficiently powered to lead to strong conclusions, but it would be helpful to add a subgroup analysis to get an indication of whether the main results are similar across different groups or driven by one group in particular.

Response:

Thank you for your feedback. In response to your advice, we divided the participants into two groups (walking aid user or not) and reanalyzed the %MoS (lines 207–212). Both groups had the same tendency, and there were significant differences only in walking aid users between when the paretic leg led and the non-paretic leg led. Thus, we have added this finding to the Results (lines 270–276).

Further minor issues need to be considered:

• The abstract is confusing when read without prior knowledge of the paper. For instance, terms as ‘first foot off’ should be more clearly described as well as a % margin of stability of over 100% (to be normalized to 100% implies that 100% is the maximum value), also it is not mentioned in which direction (medial or lateral) these values are directed.

Response:

Thank you for your suggestion. We have clarified that we investigated mediolateral dynamic stability by focusing on first foot off and the first initial contact of the leading limb in the Abstract (lines 20–21). Additionally, the direction of the positive margin of the stability value was specified (lines 26–29).

• Line 39, posture is not always unconsciously controlled during walking and while it is true that gait initiation needs to be done voluntary, it could be debated to what extend posture is consciously controlled in this task.

Response:

Thank you for your suggestion. We have added other previous studies indicating that brain activity precedes GI and argue that there are changes in muscle activity in GI in patients with stroke (lines 42–46).

• Line 45-47. “Thus, physical …” to “… with stroke.” This statement does not flow well. Why does a physical therapist determine this? Should patient not decide this?

Response:

Thank you for your suggestion. We have changed the sentence to “physical therapists sometimes need to confirm that individuals have selected an effective leading limb, especially individuals with stroke.” (lines 53–54).

• Line 59-60. Displacement needs to be defined better. Displacement cannot be defined as instability as the CoP is always moving, without always being instable. Furthermore, this neglects the functional role of CoP fluctuations (e.g. van Emmerik; van Wegen, doi:10.1097/00003677-200210000-00007).

Response:

Thank you for your feedback. As you pointed out, “displacement of CoP” is an ambiguous definition. Accordingly, we have changed the term to “range of CoP movement” and defined it as the “peak to peak amplitude of the CoP movement” (lines 75–79).

• Line 66-67 & 75. Why was the number of 38 individuals chosen? Was a power calculation done based on the mentioned (but underpowered) previous studies?

Response:

Thank you for your question. Because no previous study has analyzed the MoS during GI in individuals with stroke, which is the main outcome in this study, we could not estimate the sample size.

• Line 85-107. How was a loss of balance prevented in this study? Did the participants wear a harness or was there a therapist nearby?

Response:

Thank you for your suggestion. Guarded assistance during the task prevented the participants from falling. The participants did not wear a harness in this study. We have added this information to the Study protocol section (lines 135–136).

• Line 154-158. When discussing for how many participants %MoS exceeded 100%, was this measured on the two instances %MoS was sampled or the absolute minimal %MoS value for a participant?

Response:

Thank you for your suggestion. The number of participants who had a %MoS that exceeded 100% was determined by the absolute minimum %MoS value. We have added the explanation “the absolute minimum %MoS value exceeded 100% in 20 of the 38 participants” (lines 239 and 241–242).

• Line 158. “No participants had negative %MoS values in the mediolateral direction.” A negative %MoS would very quickly lead to a loss of stability and/or fall, which I assume would be prevented in the experimental design, to avoid injuries to the participant. Was it even possible to score a negative %MoS?

Response:

Thank you for your suggestion. We previously identified a negative MoS in the lateral direction during gait measurement, such as near-falls (Osada et al.; doi: 10.1016/j.arrct.2021.100156). The participants were stopped by manual assistance to prevent falling immediately after a negative MoS value. Because there were no such near-falls in this study, we stated that “No participants had negative %MoS values in the mediolateral direction.”

• Line 173. No hypothesis has been stated in the introduction.

Response:

Thank you for your suggestion. Our hypothesis has been added to the end of the Introduction (lines 92–96).

• Figure 3. The standard deviation lines are not quite clear because of the added shading and their overlap. I would suggest taking away the shading and using a dotted line around the dashed line for the paretic leg and a fine solid line to indicate the standard deviation for the non-paretic leg.

Response:

We agree with this comment. We have modified Figure 3 in line with your suggestion.

Finally, I have highlighted elsewhere in the review system that I believe more could be done to adhere to the Data Availability policy of PLOS. I would like to further clarify this point here. I believe this is a very relevant study which collected data in a complex setting from a very select group. Sharing the raw data would not only increase the value of this article (improve reproducibility and allow for external validation), it would also hold ethical benefits as it would allow others to perform pilot analyses, without using valuable time and effort of this vulnerable patient group.

Response:

Thank you for your suggestion. We have added all relevant data within the paper and its Supporting Information files.

Reviewer #2:

We wish to express our deep appreciation for your feedback. As indicated in our point-by-point responses below, we have taken all of your comments and suggestions into consideration in the revised manuscript. Major revisions are highlighted in yellow in the marked-up copy of the manuscript.

General Comments

1. The authors’ intended meaning of dynamic stability is not clearly defined and I believe is a point of confusion throughout the manuscript as they also mention balance control, particularly in the discussion. Being in a stable position is not the same as being able to control that position. Whether the paretic or non-paretic limb should be used to initiate gait would be dependent on a combination of these factors: in which configuration (paretic vs. non-paretic leading limb) are they most stable and in which configuration do they have the best ability to control their balance? This important distinction and consideration is not addressed by the authors, which I believe leads to the contradictions noted in the discussion (see next comment).

Response:

Thank you for your suggestion. We have clearly defined dynamic stability as “only a %MoS between 0 and 100 would indicate stability in single support” (lines 192–195) and modified the Discussion (lines 290–293, 315–322, 346–349).

2. In general, the discussion tends to contradict the authors’ conclusion more than it supports their conclusion (see specific comments 16-18). This is a major concern which suggests that the authors are not drawing appropriate conclusions from their data.

Response:

Thank you for your suggestion. In response to your specific comments 16–18, we have improved the consistency of the Discussion by modifying the interpretation of the %MoS.

3. The methods mention the subjects had a favored leading limb and that this limb was always used first in the task. However, it is not clarified whether participants “favored” their paretic or non-paretic limb and how this might have affected their findings. How did results differ between the favored and unfavored limb as compared to the paretic and non-paretic limb? Which limb did the participants select and how did this compare to their biomechanics based on your metrics?

Response:

Thank you for your question. Of the 38 participants, 17 favored the use of the paretic leg as the leading limb. We have added this number to the beginning of the Results section (line 229). However, we believe that the article will be complicated and confusing if we add an analysis of the preferences of the leading limbs. Therefore, we have stated that it is necessary to analyze the biomechanics based on the leading limb preference in the Study limitations (lines 360–361).

4. Please put citations inside punctuation. This greatly improves readability of the manuscript.

Response:

Thank you for your suggestion. We have placed the citations inside punctuation.

Specific Comments

Introduction

1. Line 39: The authors state that posture is unconsciously controlled during walking; however, it would be more accurate to state that posture is unconsciously controlled during “steady-state” or “unperturbed” walking.

Response:

Thank you for your suggestion. We have changed “Posture is unconsciously controlled during walking” to “Posture is unconsciously controlled during steady-state walking” (line 41).

2. Lines 49-51: The interpretation of the findings of reference 5 seem to be overstated by the authors. While the article does suggest that the use of the paretic leg as the leading leg is associated with difficulties activating the tibialis anterior and gluteus medius muscles during the anticipatory postural adjustment phase of gait initiation, the same article also states “In contrast, the use of the paretic leg as the trailing leg seems to challenge balance to a greater extent during GI” and in their conclusion “GI is facilitated when the non-paretic leg is used as the trailing leg because the weakness of the paretic leg leads to difficulties in supporting body weight during the upcoming stance phase.” Please accurately present the findings of the previous study in the introduction.

Response:

To address this comment, we have added the following findings of the previous study to the Introduction: “However, the same article also states that use of the non-paretic leg as the leading limb seems to challenge balance to a greater extent during GI because the weakness of the paretic leg leads to difficulties in supporting body weight during the upcoming stance phase.” (lines 59–62).

3. Lines 54-57: This sentence requires some clarification. First, what were the different evaluation indices used in each study? Second, it is unclear what the conclusions of references 7 and 8 (Tokuno and Brunt, respectively) found with regards to which leading limb is preferable.

Response:

To address this comment, we have added a clarification of each previous study. Hesse et al. compared the change in CoP, stride, and duration in 14 participants with stroke and recommended use of the paretic leading limb. Tokuno and Eng compared the ground reaction forces and GI speed in 13 participants with stroke and recommended use of the non-paretic leading limb. A previous study by Brunt et al. has been removed because it does not mention the superiority of the leading limb selection during GI (lines 63–70).

4. Line 61: Statement should read “However, center of pressure displacement…”

Response:

Thank you for this suggestion. We have added “displacement” to the sentence (line 77).

5. Lines 62-64: Please elaborate on why the use of the extrapolated center of mass/margin of stability is a superior measure of dynamic balance than center of pressure displacement.

Response:

Thank you for raising this point. We have added an explanation of why the use of center of pressure is not better than that of extrapolated center of mass (lines 77–81).

6. Lines 67-69: Please more clearly define the purpose statement. As written, the purpose is unnecessarily wordy.

Response:

Thank you for your suggestion. We have clarified the purpose of this study (lines 88–90).

Methods

7. Lines 82-83: Please explicitly state the university and hospital which gave approval for this study. Also, this sentence does not need to be its own paragraph.

Response:

Thank you for your suggestion. We have added the university and hospital name (lines 109–110).

8. Lines 96-98: Several issues need to be addressed: What is meant by “favored” leading limb and how was this determined? Did the participant’s preference of favored limb influence any of the outcome measures? What are the potential limitations of not randomizing the initial leading limb? Was the initial stance width controlled? How might the initial stance width influence the MOS?

Response:

Thank you for your suggestion. We have added the reason why the leading limb was not determined randomly (lines 133–135). Eventually, we obtained three trials for each task, without randomization of the leading limb selection. This is important because random determination of the leading limb tended to cause the patient to freeze and behave unnaturally. Therefore, first, participants repeated the trial without an instruction on which limb to be used as the leading limb, and then, participants were instructed to use the other leading limb for three more trials. 

The initial stance width was controlled with the pelvis width because the wider the initial stance width, the greater the mediolateral movement (line 124).

9. Lines 103-104: Given the importance of accurately identifying the center of mass location in your primary outcome measure, the calculation of the center of mass location should be more fully explained.

Response:

Thank you for your suggestion. We have added the details of how to calculate the center of mass. According to a previous study (Dempster, 1955), center of mass was calculated using anthropometric data for the following 12 link-segment model: foot (2), lower leg (2), thigh (2), pelvis, upper trunk, upper arm (2), and lower arm (2) (lines 145–153).

10. Lines 127-128: What is the purpose of the “additional indices” used in the study? Why were these three selected? What additional information do they provide?

Response:

Thank you for your suggestion. The additional indices were used to compare the advantages and disadvantages of leading limb selection with the results of a previous study. In each previous study, these indicators were used to state the advantage of the leading limb selection (lines 196–197, 200–202).

11. Lines 128-132: It would be helpful to number the 3 phases (e.g., 1) the postural phase, …; 2) the monopodal phase…).

Response:

We agree with your suggestion. We have added the number to the three phases (lines 159–161, 199–200).

12. Lines 134-135: The calculation of the lateral pelvic tilt angle is not sufficiently explained. What markers were used?

Response:

Thank you for your suggestion. We have added the details of the attached marker points (lines 149–150).

13. In Figure 3, the time is defined from 0-200%; however, nowhere in the methods is the time normalization of the data explained. Please define these methods.

Response:

Thank you for this suggestion. We have changed Figure 3 from 200% normalized to 100% normalized to match the figure legends.

Results

13. While p-values/effect sizes are presented in Table 2, it would be helpful to have these data included in the text. Moreover, the authors should provide a metric of how much larger/smaller (raw difference or percent difference between averages/medians) a variable was for a given condition when describing their results.

Response:

Thank you for your suggestion. We have added the p-values, effect size, and raw difference between medians to the Results section (lines 261, 267, 270).

Discussion

14. Lines 183-184: In which condition did the %MoS exceed 100% in more than half of the participants? Please clarify.

Response:

Thank you for your question. We have clarified the conditions under which the %MoS exceeded 100% (lines 303–304).

15. Lines 184-186: The statement about negative behaviors associated with paretic limb loading seem to contradict the authors’ assertion that the non-paretic leading limb (paretic trailing-limb) is better.

16. Lines 188-190: The statement that a large MOS in the lateral direction indicated poor ability to maintain balance appears to contradict the author’s assertion that the non-paretic leading limb is better based on their findings that a larger MOS was observed with the non-paretic leading limb.

17. Lines 203-204: Again, this statement that individuals with stroke may find it easier to used the paretic limb as the leading limb contradicts their conclusion.

Response:

Thank you for your suggestions (15–17). We have changed the interpretation of the results from “higher %MoS value indicate good stability” to “a %MoS exceeding 100% suggests instability.” Therefore, the conclusion was changed to “use of the non-paretic leading limb results in lower dynamic stability in the medial direction during the first step. However, there is higher dynamic stability in the lateral direction than with use of the paretic leading limb. Patients with stroke may intentionally maintain such an excessive %MoS to avoid lateral perturbation, which leads to falls.” (lines 33–38, 290–293, 315–322, 346–357, 375–377).

Tables and Figures

1. Figure 3: The standard deviation of the non-paretic limb is hard to distinguish from that of the paretic limb (it’s unclear where the lower bound of the standard deviation of the non-paretic limb is).

Response:

Thank you for your suggestion. We have modified Figure 3 to delete the overlapping shading.

---

## [Decision Letter · Decision Letter 1]

23 Mar 2022

PONE-D-21-29688R1Differences in mediolateral dynamic stability during gait initiation according to whether the non-paretic or paretic leg is used as the leading limbPLOS ONE

Dear Dr. Osada,

Thank you for submitting your manuscript to PLOS ONE. After careful consideration, we feel that it has merit but does not fully meet PLOS ONE’s publication criteria as it currently stands. Therefore, we invite you to submit a revised version of the manuscript that addresses the points raised during the review process.

Please address the remining minor recommendations.

We look forward to receiving your revised manuscript.

Kind regards,

Peter Andreas Federolf

Academic Editor

PLOS ONE

Journal Requirements:

Reviewers' comments:

Reviewer's Responses to Questions

**Comments to the Author**

1. If the authors have adequately addressed your comments raised in a previous round of review and you feel that this manuscript is now acceptable for publication, you may indicate that here to bypass the “Comments to the Author” section, enter your conflict of interest statement in the “Confidential to Editor” section, and submit your "Accept" recommendation.

Reviewer #1: All comments have been addressed

Reviewer #3: (No Response)

2. Is the manuscript technically sound, and do the data support the conclusions?

Reviewer #1: Yes

Reviewer #3: Yes

3. Has the statistical analysis been performed appropriately and rigorously? 

Reviewer #1: Yes

Reviewer #3: Yes

4. Have the authors made all data underlying the findings in their manuscript fully available?

Reviewer #1: Yes

Reviewer #3: Yes

5. Is the manuscript presented in an intelligible fashion and written in standard English?

Reviewer #1: Yes

Reviewer #3: Yes

6. Review Comments to the Author

Reviewer #1: (No Response)

Reviewer #3: This study examines mediolateral stability during gait initiation post-stroke depending on whether participants use the paretic or nonparetic leg as the leading limb. While previous work has investigated the selection of a leading limb, results have been inconsistent depending on the outcome measures and have generally not focused on balance control. A strength of this study is the large sample size of 38 participants. The authors found that when the nonparetic leg was used as the leading limb, the MoS indicated greater instability in the medial direction. This result likely reflects insufficient weight transfer to the paretic leg during nonparetic foot off. Issues with the interpretation of results have been rectified according to previous review comments. However, I have a few comments to improve the quality of this paper.

It would be useful to also include results of each subgroup (self-selected paretic leading limb, nonparetic leading limb, mixed leading limb) separately and discuss whether the self-selected leading limb resulted in better balance. At the very least, information on which limb (s) were selected in the three trials with no instruction should be included in the supplementary files. Moreover, the results from each of the three trials for each condition should be included in the supplementary data, not just the average.

This paper is intelligible but it would benefit from additional editing for conciseness and clarity, preferably by a native English speaker.

Line 51 – 54: A citation is needed for this sentence describing the role of the trailing limb, or clarify that the citation is also [7].

Lines 75 – 76: Citations are needed for “COP movement is the classical method for analyzing movement stability.” To my knowledge, COP movement is used most often for postural stability, not dynamic balance during walking.

Lines 75-81: Most of this paragraph justifying the use of MoS over CoP displacement could be removed. MoS is a well-established measurement for dynamic balance and it is clear that MoS is a better measure than CoP movement for this gait initiation task.

Line 84: While MoS is a common measurement of dynamic balance, I would refrain from calling it a “gold standard” as there are other measures of balance (such as whole body angular momentum) provide more insight into the underlying causes of balance deficits and are better correlated to clinical test scores, as described in [32]: Vistamehr, A., Kautz, S.A., Bowden, M.G. and Neptune, R.R. (2016). Correlations between measures of dynamic balance in individuals with post-stroke hemiparesis. Journal of Biomechanics 49(3): 369-400.

Lines 92 – 96: More justification for the hypothesis is needed other than “another study found this”. What are the underlying reasons that the nonparetic leading limb could be more stable?

Line 294 – 332: This is an important but very long paragraph. It should be edited for conciseness and/or divided into shorter paragraphs.

Lines 333 - 345: A discussion of why the pelvic tilt is higher when the paretic leg is the leading limb would be useful. Perhaps hip hiking to lift the paretic leg.

7. PLOS authors have the option to publish the peer review history of their article (what does this mean?). If published, this will include your full peer review and any attached files.

Reviewer #1: No

Reviewer #3: No

---

## [Author Response · Author response to Decision Letter 1]

8 Apr 2022

Detailed Response to Reviewers.

Reviewer #3:

We wish to express our deep appreciation for your feedback. As indicated in our point-by-point responses below, we have taken all of your comments and suggestions into consideration in the revised manuscript. Major revisions are highlighted in yellow in the marked-up copy of the manuscript.

Comments

It would be useful to also include results of each subgroup (self-selected paretic leading limb, nonparetic leading limb, mixed leading limb) separately and discuss whether the self-selected leading limb resulted in better balance. At the very least, information on which limb (s) were selected in the three trials with no instruction should be included in the supplementary files. Moreover, the results from each of the three trials for each condition should be included in the supplementary data, not just the average.

This paper is intelligible but it would benefit from additional editing for conciseness and clarity, preferably by a native English speaker.

Response:

Thank you for this feedback. In line with your advice, we performed further statistical calculations on the difference in the leading limb between tasks with and without instruction. However, there was no significant difference between the two. The statistical results are shown in the table below. To avoid making the study overly complex, the additional analysis you suggested has not been added to the main text. The information on which limb was selected in the three trials with or without instruction and raw values before averaging have been added in the supplementary files. 

The manuscript has been professionally edited by a native speaker familiar with this area of research. We have attached a certificate of editing.

 Leading limb without instruction Leading limb with instruction p-value Effect size (r)

%MoS (%) 

%MoS-FO#, median (IQR) 95.1 (22.2) 95.9 (39.6) 0.420 -0.13

%MoS-IC#, median (IQR) 160.0 (28.1) 159.8 (44.3) 0.856 -0.03

#Non-normally distributed; %MoS-FO, %MoS at first foot off; %MoS-IC, %MoS at first initial contact; IQR, interquartile range; MoS, margin of stability.

Line 51 – 54: A citation is needed for this sentence describing the role of the trailing limb, or clarify that the citation is also [7].

Response:

Thank you for noting this. We have clarified the citation in the sentence describing the role of the trailing limb (line 52).

Lines 75 – 76: Citations are needed for “COP movement is the classical method for analyzing movement stability.” To my knowledge, COP movement is used most often for postural stability, not dynamic balance during walking.

Response:

We have changed the sentence from “analyzing movement stability” to “analyzing postural stability” and added citations (lines 75-76).

Lines 75-81: Most of this paragraph justifying the use of MoS over CoP displacement could be removed. MoS is a well-established measurement for dynamic balance and it is clear that MoS is a better measure than CoP movement for this gait initiation task.

Response:

Thank you for this suggestion. We have removed the sentences justifying the use of MoS instead of CoP (lines 76-78).

Line 84: While MoS is a common measurement of dynamic balance, I would refrain from calling it a “gold standard” as there are other measures of balance (such as whole body angular momentum) provide more insight into the underlying causes of balance deficits and are better correlated to clinical test scores, as described in [32]: Vistamehr, A., Kautz, S.A., Bowden, M.G. and Neptune, R.R. (2016). Correlations between measures of dynamic balance in individuals with post-stroke hemiparesis. Journal of Biomechanics 49(3): 369-400.

Response:

We agree with this suggestion and have deleted the phrase “gold standard” (lines 81-82).

Lines 92 – 96: More justification for the hypothesis is needed other than “another study found this”. What are the underlying reasons that the nonparetic leading limb could be more stable?

Response:

To address this comment, we have added the rationale for our hypothesis (lines 89-92).

Line 294 – 332: This is an important but very long paragraph. It should be edited for conciseness and/or divided into shorter paragraphs.

Response:

Thank you for this feedback. We have divided the long paragraph into three parts.

Lines 333 - 345: A discussion of why the pelvic tilt is higher when the paretic leg is the leading limb would be useful. Perhaps hip hiking to lift the paretic leg.

Response:

Thank you for this suggestion. We have added discussion noting that the excessive pelvic tilt is compensatory hip hiking for toe clearance (lines 337-338).

---

## [Editor Report · Decision Letter 2]

12 Apr 2022

Differences in mediolateral dynamic stability during gait initiation according to whether the non-paretic or paretic leg is used as the leading limb

PONE-D-21-29688R2

Dear Dr. Osada,

We’re pleased to inform you that your manuscript has been judged scientifically suitable for publication and will be formally accepted for publication once it meets all outstanding technical requirements.

Kind regards,

Peter Andreas Federolf

Academic Editor

PLOS ONE

---

## [Editor Report · Acceptance letter]

18 Apr 2022

PONE-D-21-29688R2 

Differences in mediolateral dynamic stability during gait initiation according to whether the non-paretic or paretic leg is used as the leading limb 

Dear Dr. Osada:

I'm pleased to inform you that your manuscript has been deemed suitable for publication in PLOS ONE. Congratulations! Your manuscript is now with our production department. 

Kind regards, 

on behalf of

Dr. Peter Andreas Federolf 

Academic Editor

PLOS ONE